# Multiple Roles of m6A RNA Modification in Translational Regulation in Cancer

**DOI:** 10.3390/ijms23168971

**Published:** 2022-08-11

**Authors:** Guillermo Fernandez Rodriguez, Bianca Cesaro, Alessandro Fatica

**Affiliations:** Department of Biology and Biotechnology ‘Charles Darwin’, Sapienza University of Rome, 00185 Rome, Italy

**Keywords:** translation, m^6^A, cancer

## Abstract

Despite its discovery in the early 1970s, m^6^A modification within mRNA molecules has only powerfully entered the oncology field in recent years. This chemical modification can control all aspects of the maturation of mRNAs, both in the nucleus and in the cytoplasm. Thus, the alteration in expression levels of writers, erasers, and readers may significantly contribute to the alteration of gene expression observed in cancer. In particular, the activation of oncogenic pathways can lead to an alteration of the global rate of mRNA translation or the selective translation of specific mRNAs. In both cases, m^6^A can play an important role. In this review, we highlight the role of m^6^A in the regulation of translation by focusing on regulatory mechanisms and cancer-related functions of this novel but still controversial field.

## 1. Introduction

Translational regulation plays a central role in the control of gene expression, and it is often dysregulated in cancer. In particular, oncogenic pathways may act by regulating the global rate of mRNA translation or the selective translation of specific mRNAs [1]. mRNA translation is a complex process divided into four steps: Initiation, elongation, termination, and ribosome recycling. Generally, mechanisms that regulate gene expression at the translational level act on translation initiation [2]. Eukaryotic mRNAs contain the cap structure at the 5′-end, which is constituted by an N^7^-methylguanosine (m^7^G) connected via a 5′ to 5′ triphosphate bond to the first transcribed nucleotide [3]. The cap structure plays a critical role in translation initiation and is recognized by the eukaryotic translation initiation factor (eIF) 4F, which contains the cap-binding protein eIF4E, the scaffold protein eIF4G, and the RNA helicase eIF4A. eIF4G also interacts with the poly(A)-binding protein (PABP), which promotes mRNA circularization and reinitiation efficiency of ribosomes after translation termination. The AUG start codon is recognized through a scanning mechanism by the 43S pre-initiation complex (PIC) that is recruited to the cap structure by eIF4F. The PIC is formed by the 40S small ribosomal subunit, different eukaryotic initiation factors (eIF1, eIF1A, eIF2, eIF3, eIF5), and the ternary complex (TC), composed of eIF2, initiator methionyl tRNA, and GTP [2]. AUG start codon recognition by PIC induces GTP hydrolysis in the ternary complex, the release of the initiation factors, and the joining of the large 60S ribosomal subunit to form the 80S initiation complex. The 80S complex is then ready to synthesize the peptide chain [2]. A single mRNA may contain multiple 80S ribosomes and it is referred to as a polyribosome or polysome. The higher the number of ribosomes, the higher the translation rate of the mRNA. The initiation of viral mRNAs and some cellular mRNAs can be cap-independent and rely on internal RNA structures, referred to as the internal ribosome entry site (IRES), which directly recruits the PIC on the AUG start codon [2] or an alternative cap-recognition mechanism mediated by eIF3d [4].

In addition to the m^7^G, mRNAs may also contain different internal chemical modifications. In mammalian cells, the first and second transcribed nucleotides are 2′-O-methylated on the ribose moiety [5]. Moreover, if the first nucleotide is A, it can also be methylated at the N^6^-position (m^6^A_m_) [6]. 2′-O ribose methylations do not affect mRNA expression but are important to discriminate between self and non-self RNA, while the role of m^6^A_m_ in mRNA regulation and particularly in translation is still controversial. However, the most abundant internal modification is the N^6^-methyladenosine (m^6^A) [7]. m^6^A levels in mRNA are often deregulated in cancer and, more importantly, it is the only internal modification in mRNA with an established role in translation regulation.

In this review, we describe the expanding roles of translation regulation by m^6^A modification in cancer. In particular, we highlight regulatory mechanisms and cancer-related functions of this novel but still controversial field.

## 2. An Overview of m^6^A Regulators

The methyltransferase complex responsible for the vast majority of m^6^A modifications in mRNAs is composed of METTL3/METTL14 proteins, where METTL3 is the catalytic subunit and METTL14 is required for RNA binding. The complex recognizes and modifies A within the DRACH motif (D = A, G, U; R = A, G; H = A, C, U) during RNA transcription (reviewed in [7]). However, approximately 20% of DRACH motifs are methylated. m6A modifications are enriched in the terminal exon, near the STOP codon or in the 3′-untranslated region (3′-UTR). The U6 snRNA methyltransferase METTL16 is also involved in the modification of m^6^A sites in a small number of mRNAs and non-coding RNAs but it recognizes a different RNA sequence [8]. m^6^A modification can be removed by ALKBH5 (alkB homolog 5) and FTO (fat mass and obesity-associated protein) demethylases [7].

Even if m^6^A per se can impact the local RNA structure by altering the Hogsteen base-pairing, its effects on mRNA expression are generally mediated by specific protein readers. The YTH protein domain family are the only proteins that specifically recognize m^6^A modification, independently from the RNA sequence. In mammals, there are five YTH readers: YTHDC1, YTHDC2, and the paralogs YTHDF1, YTHDF2, and YTHDF3. YTHDC1 is the only nuclear reader, and it is involved in the regulation of nuclear processes such as transcription, splicing, and RNA export [9]. YTHDC2 is an RNA helicase that specifically acts during gametogenesis by degrading mRNAs. However, its function has been recently shown to be independent of the m^6^A binding domain [10,11]. Notably, several organisms have YTHDC2 orthologs in which the YTH is not present [12], thus indicating that YTHDC2 evolved to function in an m^6^A-independent manner. Cytoplasmic mRNA regulation by m^6^A is mainly controlled by the YTHDFs paralogs YTHDF1, YTHDF2, and YTHDF3. Although these proteins show high amino acid identity and equivalent binding sites in the transcriptome [9], different functions were initially ascribed to individual YTHDF proteins: YTHDF1 stimulates mRNA translation [13], YTHDF2 promotes mRNA degradation [14], and YTHDF3 has both functions [15]. However, recent studies reported that YTHDF proteins function together in a redundant manner only on the degradation of m^6^A-containing mRNAs [16,17]. At present, the action of YTHDF proteins is still controversial.

m^6^A regulators are often deregulated in cancer where they can play both oncogenic and oncosuppressive roles [18]. In the following sections, we focus on the specific role of m^6^A in translation regulation and its impact on cancer.

## 3. Regulation of mRNA Translation by m^6^A

The influence of m^6^A modifications on translation can be both positive and negative. There are currently many different, often controversial models. The position of m^6^A in mRNA influences the functional impact of its translation regulation. Indeed, proteins that bind within the coding region can be removed by ribosomes during translation elongation. Thus, m^6^A readers proteins bound on 5′- and 3′- untranslated regions (5′- and 3′- UTRs) will mediate cytoplasmic effects, while m^6^A modifications in coding regions will mainly affect the binding of tRNAs during the elongation process. Therein, the latter is generally independent of readers. In several cancer types, m^6^A modification was shown to be required for maintaining the high translational rate of oncogenic proteins [18].

### 3.1. Translational Regulation by m^6^A in 5′-UTRs

Although rare, m^6^A sites were identified in the 5′-UTR of different mRNAs. Furthermore, they were found to change in response to stress [19]. m^6^A in the 5′-UTR stimulates cap-independent translation by recruitment of the initiation factor eIF3 [20] (Figure 1a). This mechanism was initially demonstrated to be responsible for the induction of hsp70 (heat shock protein 70) upon heat shock [19,20]. In this condition, the YTHDF2 reader translocates in the nucleus protecting specific mRNAs from FTO-mediated demethylation, among which hsp70 mRNA, thereby enabling their cap-independent translation. However, the study did not clarify how these transcripts are selected by nuclear YTHDF2, which recognized only m^6^A without specificity of the sequence [9], as well as the mechanisms that translocate YTHDF2 in the nucleus without affecting the other two paralogs YTHDF1 and YTHDF3. A later study showed that the m^6^A-dependent translation of hsp70 requires the ABCF1 protein (also known as ABC50) [21]. ABCF1 belongs to the ATP-binding cassette (ABC) transporter family but lacks the transmembrane domains, which are characteristic of most ABC transporters and are regulators of translation initiation [22]. Conversely to the initial model, in which hsp70 translation was shown to depend on direct recognition of m^6^A by eIF3, ABCF1 stimulates cap-independent translation by interacting with the eIF4G and the TC component eIF2 in a stress-dependent manner [21] (Figure 1b). Notably, the m^6^A methyltransferase METTL3 was also found to be translated by ABCF1 in a positive feedback loop that could act when the cap-dependent translation is inhibited [21]. Surprisingly, ABCF1 also stimulates the translation of 5′-UTR modified mRNAs under normal growth conditions [21], thus indicating the existence of a cap-independent mechanism that relies on m^6^A modification even in the absence of stress. The same study showed that the YTHDF3 reader, but not YTHDF1 and YTHDF2, was required for cap-independent translation. However, it is not clear how ABCF1 is recruited to m^6^A modifications in the 5′-UTR region and whether YTHDF3 is involved in its recruitment. Further studies are needed to understand how specific m^6^A sites are maintained in the 5′-UTR of specific mRNAs.

M^6^A modification in the 5′-UTR was also shown to regulate the translation of ATF4 (Activating Transcription Factor 4) during the integrated stress response (ISR) [23] (Figure 1c). ISR is a prosurvival pathway that is initiated in response to different extrinsic and intrinsic factors [24]. ATF4 translation is required for the expression of stress-responsive genes [24]. ISR induction results in the phosphorylation of eIF2 with a consequent reduction in TC levels and a decrease in translation initiation efficiency. ATF4 mRNA contains two upstream open reading frames (uORFs) that precede the ATF4 coding region, uORF1 and uORF2. uORF2 overlaps with the ATF4 coding region but with a different reading frame. In normal conditions, with a high level of TC, initiation of the translation from uORFs is favored over ATF4 translation. Moreover, uORF2 start codon selection also depends on the presence of m^6^A modifications in the 5′-UTR of ATF mRNA. Upon stress, ATF4 translation is promoted by the concomitant decrease in TC and demethylation of m^6^A sites by ALKBH5. Global analysis of m^6^A levels upon nutrient deprivation showed that different transcripts lost m^6^A modifications in 5′-UTR, thus indicating that this can be a general mechanism for translation initiation by m^6^A from the non-canonical start codon in response to stress. However, the translational regulation by m^6^A, in this case, is cap-dependent and m^6^A in the 5′-UTR is strictly required for retaining initiating ribosomes in a transcript with multiple start sites. The mechanism responsible for this regulation has not yet been elucidated, but it has been suggested that the m^6^A can be read by specific RNA binding proteins that will, in turn, decrease the ribosome scanning efficiency [23]. Interestingly, ISR plays an important role in cancer, and its activation is required for tumor cell survival under stress and resistance to therapy. Therein, this indicates that the modulation of m^6^A levels can be utilized as a novel strategy to reduce ISR activation in tumors.

In tumors, the regulation of translation by m^6^A sites in the 5′-UTR was initially demonstrated in colorectal cancer [25]. Here, the YTHDC2 RNA helicase promotes the translation of the HIF-1α and Twist1 in hypoxia and facilitates the epithelial–mesenchymal transition (EMT), which plays a relevant role in cancer metastases. YTHDC2 binds to m^6^A sites and unwinds the RNA structures in the 5′-UTR, thus facilitating ribosome scanning during initiation [25] (Figure 1d).

The importance of m^6^A methylation within 5′-UTR in cancer was also shown in melanoma cells [26]. Melanoma cells that acquire resistance to BRAF (B-Raf Proto-Oncogene Serine/Threonine-Protein Kinase) and MEKs (Mitogen-Activated Protein Kinases) inhibitors, despite a general decrease in translation, showed increased translation of specific mRNAs that correlated with high m^6^A levels in their 5′-UTRs. These mRNAs encode for regulators of epigenetic modifications and signaling pathways that are connected to the presence of melanoma resistance cells. The effect of m^6^A modification on their translation is mediated by eIF4A helicase even if it is still not clear if it is directly recruited by m^6^A sites in the 5′-UTR [26]. This study suggests that the inhibition of METTL3 activity might be utilized as a novel strategy to overcome drug resistance in melanoma.

In breast cancer, the YTHDF3 reader was shown to be highly upregulated and bind to the m^6^A sites in 5′-UTR of its own transcript, stimulating translation in a positive feedback loop [27]. However, the molecular mechanism has not been clarified. High levels of YTHDF3 are then required for the translation of genes involved in brain metastases. A regulation depends on m^6^A sites that are not present in the 5′-UTR. Notably, the depletion of YTHDF3 is sufficient to inhibit brain metastasis and increases mice survival [27], thus indicating the lack of functional redundancy between YTHDF proteins in this context.

### 3.2. Translational Regulation by m^6^A in Coding Regions

The effect of m^6^A modifications within coding regions on mRNA translation, as well as their mechanisms of action, is still controversial. Translation elongation is conserved in all kingdoms of life, and initial work performed with bacteria showed that the presence of m^6^A within codons impairs tRNA accommodation, thus decreasing the translation elongation rate [28] (Figure 2a). Similar results were observed in HEK293T cells transfected with in vitro transcribed mRNAs containing different modifications [29]. This study showed that in human cells, m^6^A within coding regions also strongly inhibited translation, especially when it was present in the first codon position. A negative effect of m^6^A on elongation was also reported in the breast cancer cell line MCF7 in a study performed to correlate the transcription rate with translation efficiency by using reporter systems containing the luciferase gene under the control of different human promoters [30]. Additionally, in this case, the increase in m^6^A levels within the coding region of reporter transcripts produced a decrease in the translation rate. Moreover, the study also showed that the deposition of m^6^A is inversely proportional to the transcription rate. Notably, in different maize lines and during *Xenopus laevis* oogenesis, an inverse correlation between the m^6^A levels in the coding regions of mRNAs and their translational efficiency was confirmed [31,32]. The negative effect of m^6^A modification on translation elongation was also reported in mouse embryonic fibroblasts (MEF) [33], where the presence of m^6^A in highly structured regions produced ribosome pausing. Surprisingly, deleting m^6^A from these transcripts resulted in a further decrease in translation [33]. The authors of this study proposed a model in which the YTHDC2 reader, which also contains an RNA helicase domain, resolves the RNA structures containing m^6^A, therein promoting translation elongation by the ribosome (Figure 2b).

The opposite results were reported in acute myeloid leukemia (AML) cells. In this blood cell cancer, METTL3 plays an oncogenic role and is required for sustaining AML cell proliferation [34]. In this case, METTL3 is recruited to specific promoters, independently from METTL14, to install m^6^A modification within the coding region of oncogenic mRNAs during transcription. However, later crystallographic studies demonstrated that METTL14 is strictly required for METTL3 methyltransferase activity [7]. Therein, it is not clear how METTL3 can act independently from METTL14 to methylate specific mRNAs. However, methylation by METTL3 in their coding region results in an increased translation rate. The proposed mechanism relies on the reduction of ribosome stalling on m^6^A methylated GAN codons, which results in faster elongation speed [34] (Figure 2c). This peculiar mechanism has only been described in AML to date.

The importance of translation elongation regulation by m^6^A in coding regions was also demonstrated in the regulation of EMT [35]. In this case, m^6^A sites in the coding region of the Snail transcript, which encodes for a transcription factor that regulates EMT, promote Snail translation by recruiting the eukaryotic translation elongation factor (eEF) 2 through the YTHDF1 reader [35] (Figure 2d). Interestingly, the study also reported that Snail mRNA also contains several m^6^A sites in the 3′-UTR that are not required for Snail regulation. Moreover, by using cell-line-derived xenograft mice, the authors showed that the overexpression of Snail can also stimulate lung colonization by HeLa cells in the absence of the methyltransferase METTL3 [35]. Similar results were reported in breast cancer and gastrointestinal stromal tumors [36,37]. In the first case, the m^6^A deposition in keratin 7 (KRT7), an important mediator of cancer metastases, stimulates its translation via the YTHDF1/eEF1 axis [36]. In the latter, the YTHDF1/eEF1 interaction stimulates the translation of the multidrug transporter MRP1 mRNA, also known as ABCC1 (ATP Binding Cassette Subfamily C Member 1), which is involved in drug resistance [37].

### 3.3. Translational Regulation by m^6^A in 3′-UTRs

3′-UTRs contain *cis*-regulatory elements that are recognized by RNAs and proteins to regulate translation. This regulation mainly takes place at the level of the translation initiation phase. Indeed, the interaction between eIF4F and PABP brings 3′-UTR elements close to the translation initiation complex. m^6^A sites in the 3′-UTR were initially shown to stimulate mRNA circularization via the interaction between the YTHDF1 reader and the initiation factors eIF3A and eIF3B [14] (Figure 3a). Moreover, the tethering of YTHDF1 on reporter constructs was sufficient to stimulate its translation. However, it was reported that eIF3 can also directly recognize m^6^A in the 5′-UTR to stimulate the cap-independent translation (see above), so it is not clear how eIF3 would not bind to m^6^A in the 3′-UTR, which is close to the 5′-end of mRNA, independently from YTHDF1. Even if the specific effect of YTHDF paralogue proteins is currently under debate [16,17], many studies reported the importance of YTHDF1 in the regulation of translation in cancer [38,39,40,41,42,43,44,45,46,47,48,49,50,51]. Notably, a strong antitumor response was described in YTHDF1-deficient mice [39]. In dendritic cells, YTHDF1 stimulates the translation of mRNA encoding for proteases that results in efficient antigen degradation. The lack of YTHDF1 increases the presentation of tumor antigens and the activation of CD8^+^ T cells, which are required for containing tumor infiltration [39]. Therein, these results suggest that inhibition of YTHDF1 activity might be used in combination with immune checkpoint inhibitors to enhance the killing of cancer cells by the immune system.

Similarly, YTHDF3 was also shown to stimulate the translation of mRNAs containing m^6^A sites in the 3′-UTR. Initial studies indicated that YTHDF3 acts in cooperation with YTHDF1 [15,52] (Figure 3b). However, further studies indicated that YTHDF3 can also function independently from YTHDF1 [26] (Figure 3c). Interestingly, it was shown that the MYC oncogene, which is a general activator of ribosome biogenesis and translation in cancer, inhibits the translation of specific transcripts by downregulating their m^6^A levels through the transcriptional induction of the demethylase ALKBH5 [53]. Furthermore, in this case, YTHDF3 was required for their translational activation, and this effect was lost upon MYC activation [53].

### 3.4. Direct Translational Regulation by m^6^A Methyltransferases

Notably, in cancer cells, METTL3 and METTL16 methyltransferases can translocate to the cytoplasm and act as positive regulators for the translation of oncogenic m^6^A-modified mRNAs. This mechanism was initially described for METTL3 in lung cancer [54,55] and chronic myeloid leukemia (CML) [56,57]. The binding of METTL3 in the 3′-UTR of these transcripts promotes translation via interaction with the eIF3 component eIF3h [55] (Figure 4a). The tethering of METTL3 in the 3′-UTR of reported genes is sufficient to stimulate translation. Notably, the expression of a METTL3 derivative without the eIF3h interacting region is not able to induce tumors in lung cancer cell line xenograft models; thus, indicating the important role of cytoplasmic regulation of translation by METTL3 in tumorigenesis [55]. Importantly, the cytoplasmic activity of METTL3 does not require its catalytic domain. However, m^6^A installation is apparently required for METTL3 binding to mRNA in the cytoplasm. It has been proposed that cytoplasmic METTL3 does not require any readers for the recognition of m^6^A sites in the 3′-UTR. However, it is not clear how METTL3 would recognize and bind to modified transcripts in the cytoplasm because, in the nucleus, it strictly requires METTL14 for binding to consensus DRACH sequences, which do not contain m^6^A-modified adenines [7].

More recently, a similar mechanism was described for METTL16 in hepatocellular carcinoma [58]. The authors reported that METTL16 binds to m^6^A sites close to the AUG start codon and promotes translation by facilitating the interaction between eIF3a and eIF3b, and the 18 rRNA component of the 40S small subunit in the PIC [57] (Figure 4b). Conversely to METTL3, the methyltransferase domain was shown to be strictly required for the activity of METTL16 as a translational regulator and interaction with the eIF3 components. Therein, this suggests that the use of a catalytic inhibitor would inhibit both catalytic-dependent and -independent activities of METTL16. Furthermore, in hepatocellular carcinoma cells, the depletion of METTL16 produced a strong decrease in translation and impaired cell survival. Notably, METTL16 can bind more than a thousand transcripts, the m^6^A modification of which does not, however, depend on METTL16 [58].

For both METTL3 and METTL16, further studies are needed to understand the mechanism that promotes their cytoplasmic translocation and how they can discriminate and bind to specific m^6^A-modified transcripts.

### 3.5. Translational Regulation by m^6^A in Circular RNAs (circRNAs)

circRNAs are covalently closed RNA molecules produced by the back-splicing of coding transcripts (review in [59]). As many circRNAs are derived from coding exons, they may still have coding potential but without the cap structure and poly-A tail required for cap-dependent translation. Notably, many circRNAs contain m^6^A close to the AUG start codon, and a single m^6^A site is sufficient to promote circRNA translation [60,61]. m^6^A sites within circRNAs are recognized by the YTHDF3 reader that recruits the non-canonical eIF4G2 (also known as Dap5 and Nat1) translation initiation factor to stimulate cap-independent translation [60,61]. In hepatocellular carcinoma, an additional mechanism has been described for circMAP3K4, derived from the MAP3K4 gene (Mitogen-Activated Protein Kinase 4). This circRNA was found upregulated and translated into a peptide of 455 amino acids that protects cancer cells from apoptosis induced by chemotherapeutic agents [62]. In this case, circMAP3K4 translation depends on m^6^A sites that are recognized by the IGF2BP1 protein (Insulin-Like Growth Factor 2 MRNA Binding Protein 1). This is a peculiarity of circRNAs because, in linear m^6^A-modified transcripts, the binding of IGF2BP1 increases their stability [7]. Several translated circRNAs have been discovered (reviewed in [63]), and many of the produced proteins play important roles in different types of cancer [63,64,65,66,67]. Nevertheless, the mechanism of cap-independent translation has not yet been identified for most of them. Therein, the role of m^6^A in the translation of circRNAs warrants further investigation.

## 4. Conclusions

Translational regulation is crucial for cancer development [1]. Highly proliferating cells demand elevated levels of ribosomes and intense translational rates. Moreover, cancer cells need to adapt their translation under stress conditions, such as hypoxia, derived from their growth environment. Thus, they require mechanisms that integrate different steps of gene expression, from transcription to translation, to produce a general increase in protein synthesis or to enhance the translation of specific mRNAs. The greatly expanding field of RNA modifications fits precisely in this context. Among the hundreds of different modifications that RNA can undergo, the one that is most studied in the tumor field is undoubtedly m^6^A within mRNA molecules. Inhibitors against writers and erasers of m^6^A have been recently developed and have shown promising results in cancer cells and cancer mouse models [68,69,70].

However, a major problem in the field is the many different mechanisms identified for m^6^A in mRNA metabolism and translation regulation, often with opposite effects, and the lack of reproducible results between different studies. This could be due, at least partially, to the lack of specific and quantitative methods for m^6^A detection, context dependency, and the complexity of gene expression regulation by m^6^A and, eventually, its indirect effect on translation. We urgently need mechanistic studies to determine the contribution of m^6^A to translation. In this context, targeted m^6^A editors might be used to install or remove m^6^A at desired sites in cancer-related mRNAs to address the specific positional effects of m^6^A on translation. Moreover, another critical point to clarify is the redundancy between cytoplasmic readers of the YTH family and their individual contributions to translation regulation.

Nevertheless, the identification of specific translational regulations mediated by m^6^A in cancer cells might bring the development of novel therapeutic strategies. One of these might be the interference with the cytoplasmic function of METTL3 and METTL16 as positive translational regulators of oncogenes. Understanding the mechanisms and signaling pathways that drive their translocation to the cytoplasm can be useful in the design of combination therapy with drugs already used in the clinic.

The m^6^A field is relatively new and it is hoped that, in the future, methodological advancement and scientific rigor can produce definitive results on its contribution to gene expression regulation, including translation. This will surely benefit cancer studies.

## Figures and Tables

**Figure 1 ijms-23-08971-f001:**
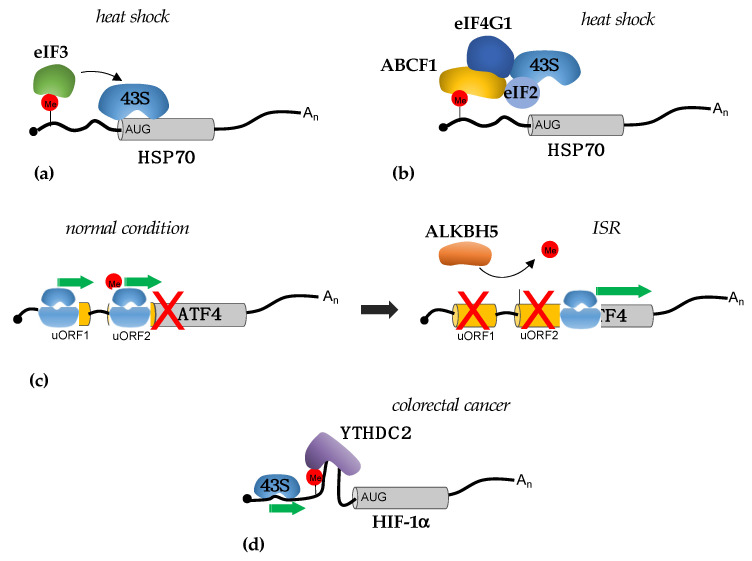
Mechanisms of translational regulation mediated by m^6^A sites in 5′-UTRs. (**a**) Proposed mechanism for the cap-independent translational regulation of Hsp70 mRNA by eIF3 during heat shock (see main text for detail); (**b**) proposed mechanism for the cap-independent translational regulation of Hsp70 mRNA by ABCF1 during heat shock (see main text for detail); (**c**) during ISR, the decrease i TC levels and loss of removal of m^6^A from uORF2 of ATF4 mRNA by ALKBH5 induces ATF translation; (**d**) in colorectal cancer cells, m^6^A sites in the 5′-UTR of HIF-1α mRNA recruit the YTHDC2 helicase that removes secondary RNA structure facilitating 43S scanning.

**Figure 2 ijms-23-08971-f002:**
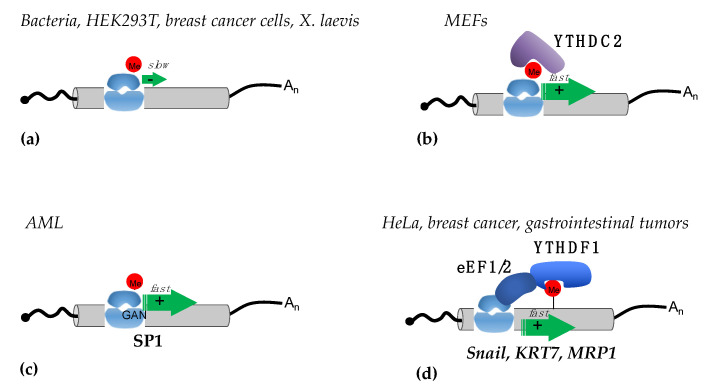
Mechanisms of translational regulation mediated by m^6^A sites in coding regions. (**a**) In bacteria, different human cell lines, and in *X. laevis,* m^6^A sites in the coding regions were shown to slow down elongation (indicated by a green arrow); (**b**) in MEFs m6A in the coding regions are recognized by the YTHDC2 reader, which by removing secondary RNA structures facilitates elongation by the ribosome; (**c**) in AML the m^6^A modifications present in GAN codons of specific mRNAs, such as SP1, increase elongation by reducing ribosome stalling; (**d**) in different tumor cell lines, m^6^A sites in the coding regions of specific mRNAs (such as Snail, KRT, and MRP1) are bound by the YTHDF1 reader, which stimulates elongation by interacting with the eEF1 and eEF2 elongation factors.

**Figure 3 ijms-23-08971-f003:**
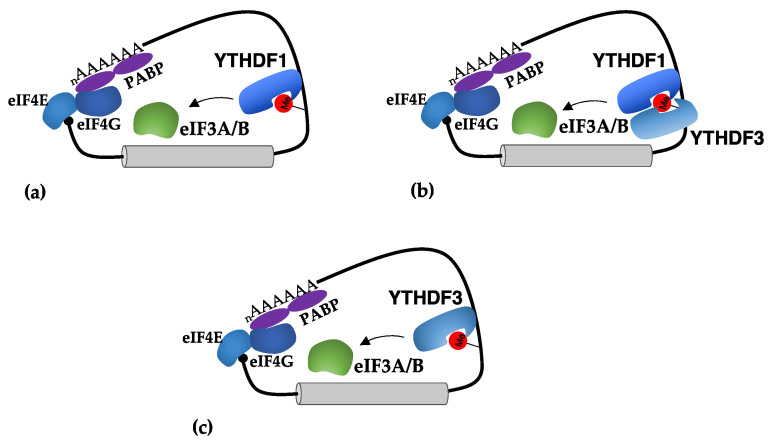
Mechanisms of translational regulation mediated by m^6^A sites in 3′-UTRs. (**a**) The binding of YTHDF1 in m^6^A sites present in the 3′-UTR stimulates translation initiation by recruiting eIF3A and eIF3B; (**b**) the YTHDF3 reader was shown to act in cooperation with YTHDF1 in stimulation of translation; (**c**) in some tumors, YTHDF3 can act independently from YTHDF1 in stimulating translation.

**Figure 4 ijms-23-08971-f004:**
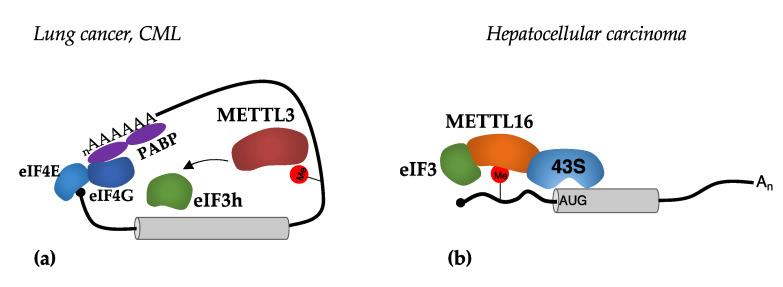
Mechanisms of translational regulation mediated by METTL3 and METTL16 methyltransferases. (**a**) The binding of METTL3 in m^6^A sites present in the 3′-UTR stimulates translation initiation by recruiting eIF3h; (**b**) the binding of METTL16 in m^6^A sites present in the 5′-UTR stimulates translation initiation by facilitating the interaction between the 43S PIC and the eIF3 translation initiation factor.

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
