# Peer review of "Multiple Roles of m6A RNA Modification in Translational Regulation in Cancer"

_ijms, 2022, doi:10.3390/ijms23168971_

Round 1
Reviewer 1 Report
The manuscript nicely summarizes the current knowledge on the functional implications of m6A modifications in cellular mRNAs in cancer. The manuscript is well organized. The authors have distinguished the implications in translation regulation of m6A modifications located in 5’-UTRs, coding region and 3’-UTRs of mRNAs always in relation with cancer. They critically analyze the current information, pointing out the controversies derived from the different works and experimental conditions used by different authors.
The manuscript is well-written.
Minor points,
1. The manuscript contains some typos and spelling errors:
Line 62 mod-ifies
Line 63 meth-ylated
Line 69 Hoogsteen
A careful checking of the entire manuscript should be performed.
2. Abbreviations. Check that all abbreviations used are explained the first time they are mentioned, e.g. TC, what does it mean?
The manuscript includes the name of many different proteins, although details of most of them are included, it is important to identify all of them; e.g. line 112 “A later study showed that m6A-dependent translation of hsp70 requires the ABCF1 protein”. No information is provided about ABCF1, readers don’t have to be familiar with those proteins and know what kind of protein ABCF1 is or what function it has. Is it a transporter, or a translation initiation factor, etc? It would also be important to know if a role in cancer has been described for ABCF1. This applies to all other proteins included.
3. Line 187 “HEK293T cells by transfecting in vitro transcribed mRNAs…” change to HEK293T cells transfected with in vitro transcribed…
4. References 33 and 56, year should be in bold.
5. Fig. 2. I want to understand that the size of the green arrow indicates the negative or positive effect, but it is not clearly indicated, In addition, the size difference and the meaning may not be obvious for all readers. May be a (+) and (-) symbol can be added, or any other modification to make it clear.
Author Response
We would like to thank the referee for the constructive comments. We have modified the text accordingly.All modifications are highlighted in the text. Here, our point-by-point response:
1. The manuscript contains some typos and spelling errors:
Line 62 mod-ifies
Line 63 meth-ylated
Line 69 Hoogsteen
A careful checking of the entire manuscript should be performed.
We have corrected the typos and spelling errors in the manuscript 2. Abbreviations. Check that all abbreviations used are explained the first time they are mentioned, e.g. TC, what does it mean?The manuscript includes the name of many different proteins, although details of most of them are included, it is important to identify all of them; e.g. line 112 “A later study showed that m6A-dependent translation of hsp70 requires the ABCF1 protein”. No information is provided about ABCF1, readers don’t have to be familiar with those proteins and know what kind of protein ABCF1 is or what function it has. Is it a transporter, or a translation initiation factor, etc? It would also be important to know if a role in cancer has been described for ABCF1. This applies to all other proteins included.
We apologize with the referee; we added the abbreviation of ternary complex in line 37 in the introduction section and checked for further mistakes. We added information about ABCF1 and added a new reference, n 22.
3. Line 187 “HEK293T cells by transfecting in vitro transcribed mRNAs…” change to HEK293T cells transfected with in vitro transcribed…
We have changed the text according to referee suggestion.
4. References 33 and 56, year should be in bold.
We modified the references.
5. Fig. 2. I want to understand that the size of the green arrow indicates the negative or positive effect, but it is not clearly indicated, In addition, the size difference and the meaning may not be obvious for all readers. May be a (+) and (-) symbol can be added, or any other modification to make it clear.
We thank the referee for the suggestion. We specified in the legend that the green arrow indicates elongation e we added (-) and (+) symbol within the arrows to indicate negative or positive effects on elongation.
Reviewer 2 Report
The manuscript by Rodriguez, Cesaro, and Fatica reviews the role of m6A RNA modification in translational regulation in cancer. The authors have done a fairly good job to review the literature and present in the form of a comprehensive review article. The m6A RNA modification has been known for a while, therefore the phenomenon is not quite new. Additionally, the role of m6A modification in cancer through the dysregulation of translational machinery is well-known despite the authors' claim that it has recently gained significance in recent years. Everything presented in the manuscript appears to be rationally justified. However, the review appears diffused. My suggestion is that the authors should shorten the review and focus only on the relevance of m6A modification in cancer. The process of m6A modification at different RNA positions distracts the reader from the role of m6A in cancer. To do that the authors may want to change sub-headings, exclude the process of modification and directly to the relevance of the modification at a specific position in cancer.
Author Response
The manuscript by Rodriguez, Cesaro, and Fatica reviews the role of m6A RNA modification in translational regulation in cancer. The authors have done a fairly good job to review the literature and present in the form of a comprehensive review article. The m6A RNA modification has been known for a while, therefore the phenomenon is not quite new. Additionally, the role of m6A modification in cancer through the dysregulation of translational machinery is well-known despite the authors' claim that it has recently gained significance in recent years. Everything presented in the manuscript appears to be rationally justified. However, the review appears diffused. My suggestion is that the authors should shorten the review and focus only on the relevance of m6A modification in cancer. The process of m6A modification at different RNA positions distracts the reader from the role of m6A in cancer. To do that the authors may want to change sub-headings, exclude the process of modification and directly to the relevance of the modification at a specific position in cancer.
We thank for the comments, but we do not agree with the referee. Even if discovered in 1970s, the importance of m6A in mRNA expression regulation was revealed only in 2012 with the development of RNA sequencing technologies to map m6A within specific mRNA molecules (see Dominissini et al. 2012. Topology of the human and mouse m6A RNA methylomes revealed by m6A-seq. Nature, 485, 201–206. http://www.nature.com/doifinder/10.1038/nature11112). Its role in regulation of translation only in 2015 (Wang et al., 2015. N(6)-methyladenosine modulates messenger RNA translation efficiency. Cell 161, 1388-99. doi: 10.1016/j.cell.2015.05.014.) followed by different mechanisms and regulators described in this review.
There are more than 200 review on the role of m6A in cancer, we do not think the IJMS readers need an additional one. On the other hand, we wanted to highlight the specific role of m6A in the regulation of translation in cancer by focusing on regulatory mechanisms, pointing out the controversies derived from the different works and experimental conditions used by different authors.